# Task–technology fit and digital textbook usage outcomes: The mediating role of positive emotion within an S–O–R framework

Ning Mi[1,2]*, Cunying Fan[1]

1 Department of College English Teaching, Qufu Normal University, Qufu, Shandong, China, 2 Center for College English Education Research, Qufu Normal University, Qufu, Shandong, China

* sdmining@qfnu.edu.cn

## Abstract

Digital textbooks are now an integral component of EFL learning environments, yet the underlying psychological mechanisms that influence digital textbook usage outcomes are still clearly unknown. This study constructs a model using the Stimulus–Organism–Response (S–O–R) theory to show how task-technology fit (TTF), which is an external stimulus variable, affects the outcomes of digital textbook usage (DTUO) by influencing the mediating role of positive emotion (POE). In this model, DTUO is defined as a second-order construct consisting of learning effectiveness (LEE) and continuance intention (COI). Using cross-sectional questionnaire data from 259 undergraduates in China, the paper employed structural equation modeling (SEM) for empirical investigation. Findings indicated that TTF positively predicted both POE and DTUO. Furthermore, POE was found to be positively associated with DTUO and played a significant mediating role in the relationship between TTF and DTUO. These findings confirm that the S-O-R theory is applicable in digital textbook learning and underscore the significance of a task-technology fit and emotional engagement. Theoretical and empirical contributions are offered for educational content developers and policymakers.

## Introduction

In the context of the rapid development of educational technology, digital textbooks have become a crucial component of contemporary classrooms, as they fulfill the demands for increased accessibility and flexibility in learning [1]. These materials are expected to enhance the quality of learning and bring long-term academic gains due to the interactive and multimedia elements involved. However, there is no assurance of sustained long-term use, and empirical evidence shows that the actual effectiveness of digital textbooks can vary dramatically, particularly in English as a Foreign Language (EFL) contexts [2,3]. Thus, identifying the precise mechanisms through

**Data availability statement:** All data are available from the Zenodo repository (DOI: 10.5281/zenodo.18632600).

**Funding:** This work was supported by the Shandong Social Sciences Planning Research Project of China (Grant No. 23CSDJ24) awarded to CY. F. The funder had no role in study design, data collection and analysis, decision to publish, or preparation of the manuscript.

**Competing interests:** The authors have declared that no competing interests exist.

which the technical potential is transformed into the successful use of digital textbooks has become an important research goal.

One of the key elements for effective technology integration is task-technology fit (TTF), which describes the extent to which the system capabilities match the needs of a specific task [4]. A range of intelligent affordances that digital textbooks provide, such as multimodal input and interactive practice opportunities, can significantly enhance perceived TTF in EFL contexts [5,6]. Previous studies primarily examine the direct effect of fit on performance, but have not focused on the effect of TTF on students' psychological states. Although educational technology interactions have explained that affective states such as enjoyment and satisfaction are crucial [7,8], their function as a psychological bridge between functional fit and usage success is insufficiently expounded.

To fill in the research gap above, the current study uses the Stimulus-Organism-Response (S-O-R) framework [9] as the theoretical lens. Within the framework, TTF is treated as the environmental stimulus (S) that activates affective states (O) of learners, which in turn influences the response (R) of the digital textbook usage. We propose positive emotion (POE) as the core "organism" in the process.

Furthermore, this study seeks to refine how digital textbook success is measured. Traditionally, learning effectiveness and behavioral intentions have been treated as isolated endpoints. Drawing on the Information Systems Success Model [10,11], we measure digital textbook learning as a consolidated outcome rather than fragmented variables. While Bhattacherjee (2001) and Venkatesh et al. (2012) emphasize that behavioral commitment (COI) is a cornerstone of technology post-adoption success [12,13], researchers in educational contexts argue that such intentions are intrinsically tied to cognitive gains and efficacy (LEE) [14]. Consequently, we operationalize digital textbook usage outcomes (DTUO) as a second-order construct that integrates both perceived learning effectiveness (LEE) and continuance intention (COI) to get learners' overall evaluative response.

In summary, proposing a mediation model grounded in the S–O–R framework, this study clarifies how task–technology fit shapes EFL learners' use outcomes through the vital mechanism of positive emotion. Accordingly, the following research questions guide this investigation:

RQ1. How does task–technology fit shape EFL learners' digital textbook usage outcomes?

RQ2. To what extent does positive emotion mediate the link between task–technology fit and digital textbook usage outcomes?

## Theoretical background and hypotheses development

### The S–O–R framework and TTF theory

Rooted in environmental psychology, the S–O–R framework establishes that environmental attributes function as stimuli (S) that shape individuals' internal affective and cognitive states (O), which in turn elicit behavioral responses (R) [9]. "Organism" component encompasses cognitive, affective, and perceptual processes mediating between stimuli and responses [15]. The S–O–R framework has exhibited robust

contextual validity, supported by extensive empirical deployment in digital behavioral studies [16,17]. For instance, it was utilized to study the effects of interactivity as a stimulus on students' MOOC continued usage as a response via the organism of engagement [18]. Similarly, Zhai et al. [19] concentrated on in what way the issue of privacy led to the development of learners' knowledge concealment cognition, which in turn affected their online cooperation based on S-O-R paradigm.

TTF is defined as how well technology features meet users' task requirements [4]. Theoretically, high TTF means that the affordances of the system provide the required practical value to perform the task, thus leading to higher utility perceptions and user satisfaction [20,21]. In the educational technology area, empirical evidence suggests that synchronization is a crucial factor to ease high cognitive load and facilitate the perceived use of digital technologies [22]. In other words, synchronization could capture the degree to which technical features such as context-aware scaffolding, interactive feedback, and multimedia integration enhance the performance of basic language learning tasks such as completing conversations [23,24]. When the technical architecture of an e-learning system reflects instructional logic, students can interact with the system more easily, and learning is more enjoyable.

The present study integrated the TTF into the S–O–R framework, so producing a unified model to assess the influence of functional fit on the digital textbook usage results of EFL learners. TTF is the outside stimulus, POE is the main organismic state, and DTUO is the response in this model. DTUO is a second-order construct that is proven by how well students think they are learning and how much they desire to keep studying. Fig 1 illustrates the proposed research model.

## Task–technology fit and digital textbook usage outcomes

The S–O–R framework posits that task–technology alignment constitutes the primary environmental stimulus (S) influencing learner response (R). TTF theory states that the features of a technological tool operate best when they suit the needs of the task the user is undertaking. This is the basis for the link between how well a task fits with technology and how well digital textbooks work [4]. Evidence suggests that increased TTF may significantly reduce cognitive friction while improving value assessments, hence affecting students' perceptions of their learning experiences in digital environments [25].

This association is particularly salient in EFL learning settings, involving sustained and repetitive practice. Wang et al. [26] emphasized the need of aligning the tool and the task when students require repetitive repetition. In the EFL contexts, digital textbooks that effectively support fundamental language learning activities are more likely to yield improved results, encompassing both cognitive instructional effectiveness and psychological commitment to continued use. Consequently, DTUO is viewed as a holistic construct that captures these multifaceted responses. From this reasoning, we propose the following hypothesis:

H1: Task–technology fit positively predicts EFL learners' digital textbook use outcomes.

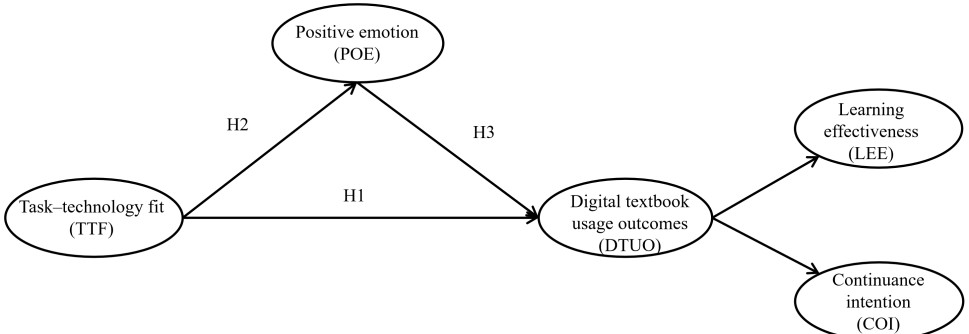

**Fig 1. The hypothesized model of the relationship between task-technology fit and digital textbook usage outcomes in the EFL learning context under consideration of positive emotion.**

## Task–technology fit and positive emotion

Task-technology fit is crucial for both how well things work and how the learner feels inside (O). It can feel amazing when the instrument and the task fit nicely together in the S–O–R framework. Students can think more clearly and quickly when the elements of a digital textbook fit the goals of the language learning assignments [27]. This nice interface makes it easier for learners to study because they don't have to figure out how to utilize the technology, naturally fostering a more positive emotional climate.

Empirical studies demonstrate that a significant degree of TTF enhances perceived usability and alleviates usage-related difficulties, hence promoting more favorable emotional experiences in technology-enhanced learning [28]. Liu et al. [29] demonstrated in a related study that the appropriate TTF can enhance emotional engagement in digital learning environments. Consequently, the following hypothesis is proposed:

H2: Task–technology fit positively predicts EFL learners' positive emotion while using digital textbook.

## Positive emotion and digital textbook usage outcomes

Positive emotions, such as enjoyment, satisfaction, and enthusiasm, indicate learners' affective states in technology-supported learning activities [30,31]. In the S–O–R model, POE is an internal psychological variable that influences learners' subsequent evaluations and behavioral tendencies toward the learning environment [9]. In digital learning contexts, learners who experience POE are more likely to sustain interest and engagement with learning activities and, thus, perceive increased instructional effectiveness [32].

In addition, previous empirical research suggests that enjoyment and interest have been found to be substantial predictors of continued long-term engagement and the intention to use digital tools [33,34]. At the same time, evidence from several domains of technology suggests that positive emotional states are strong predictors of an individual's intention to continue using a particular application [35]. As a result, positive emotion is expected to be an important variable that influences learners' overall use outcomes, including the appraisal of learning gains and the intention to continue using the technology in the long term. Accordingly, the following hypothesis is developed:

H3: Positive emotion positively predicts digital textbook usage outcomes.

## Task–technology fit, positive emotion, and digital textbook usage outcomes

In the S–O–R framework, the responses of individuals to external stimuli are believed to be primarily determined by the internal psychological states that are formed [4]. In the context of digital textbooks, task-technology fit serves as the environmental stimuli, and learners' consequent affective experience serves as the central organismic state that influences subsequent evaluations. Empirical research has shown that when technological tools fit in with the tasks, they stimulate positive emotions that facilitate the engagement of learners [36,37]. Specifically, Loderer et al. found that technology-based learning environments that fit the technical features with task requirements are more likely to trigger positive achievement emotions such as enjoyment and satisfaction, and these emotions promote deeper cognitive engagement [38].

Moreover, positive affective states are also a pivotal psychological channel through which instrumental system affordances are transformed into real educational benefits. Chao (2019) indicated that a high level of task–technology fit serves as a prerequisite for creating the emotional context of heightened satisfaction and readiness to use [39]. In the context of digital textbooks, Wang and Xing (2018) showed that task–technology fit serves as the foundation of confidence and perceived usefulness that is essential for students to reap the benefits of using learning resources [40]. Consequently, the integrated findings indicate that the task technology fit's effect on digital textbook usage success is not a direct functional consequence of fit, but rather one that is effectively conveyed through these positive affective responses. In effect,

positive emotion is a necessary mediating element that connects technical suitability with success in aspects of usage. Therefore, we hypothesize that:

H4: Positive emotion mediates the relationship between task–technology fit and digital textbook usage outcomes.

## Method

### Digital textbook

The digital version of New Standard College English (3rd edition), published by Foreign Language Teaching and Research Press (Beijing, China), was used as the instructional material in this study. This textbook is widely adopted in Chinese higher education, ensuring its representativeness in EFL contexts.

The digital version has the same core content and unit structure as the print edition while incorporating several platform-based functions, including an AI-powered tutor ("Ziyan"), smart portfolios with built-in note tools, and an automatic feedback system.

The associated digital platform supports a structured instructional sequence (motivating, enabling, and assessing). To ensure methodological consistency, all participants were required to complete the same textbook units and use the embedded digital functions as part of their regular coursework, thereby ensuring comparable exposure to the learning environment.

### Participants

A total of 259 undergraduate students enrolled in a college English course at Qufu Normal University, China, participated in this study. Participants were recruited through convenience sampling based on course registration and availability, ensuring that all respondents had direct experience with the learning context under investigation.

The adequacy of the sample size for structural equation modeling (SEM) was assessed using two widely accepted methodological criteria. First, prior methodological literature suggests that a minimum sample size of 200 is generally required to obtain reliable parameter estimates and achieve stable model convergence [41,42]. Second, sample adequacy was further evaluated with reference to the ratio of cases to observed indicators. According to recommendations [43,44] by Hair et al. (2019) and Bentler and Chou (1987), a ratio ranging from 5:1–10:1 is considered appropriate for model estimation. Given that the proposed model included 17 observed indicators, the resulting ratio of 15.2:1 exceeded these recommended thresholds. Accordingly, the sample size was deemed sufficient to support the proposed SEM-based mediation analysis with adequate statistical power.

The participants consisted of 161 females (62.2%) and 98 males (37.8%). This gender distribution is fairly typical of the university's student population. The participants were drawn from a variety of academic fields, including statistics (22.0%), history (18.5%), cybersecurity (17.4%), physics (16.6%), biology (14.7%) and psychology (10.8%).

Potential biases were considered. Firstly, because convenience sampling was conducted at a single university, generalizability to the overall undergraduate student population in China may be limited. Also, although the sample was gender imbalanced (i.e., more females), this was because it is typical of the student body at the university (i.e., teacher education institution). Secondly, although self-report measures were used, which may have introduced common method bias, we used a variety of procedural remedies, including anonymity and emphasizing that there were no right or wrong answers.

### Instrument

**Scale adaptation and validation procedure.** Research data were collected using multi-item measurement scales adapted from established instruments in the extant literature. All scales were adapted from previously validated measures and were subsequently assessed for reliability and validity in the present study to evaluate their performance in this research context.

To ensure cross-cultural equivalence and contextual relevance for Chinese undergraduates, we followed a rigorous adaptation process. First, the original English scales were translated into Chinese and then back-translated into English by two independent bilingual researchers to ensure linguistic accuracy, based on the procedure by Brislin [45]. Second, the translated items were reviewed for content and face validity by three professors with expertise in applied linguistics and educational technology. Finally, a pilot test was conducted with 30 undergraduates to evaluate the linguistic clarity and readability of the instruments. These individuals were excluded from the primary data collection. We made little changes to the language based on their feedback to make sure that all of the elements were clear and easy for the target market to understand.

Following these refinements, the final questionnaire employed a 5-point Likert scale, with 1 being "strongly disagree" and 5 being "strongly agree". Table 1 shows the final constructs, the measurement items, and their sources.

### Measures

**Task-technology fit (TTF).** The TTF scale was adapted from Gerhart et al. and Kim et al. to assess students' perceptions of how well the digital textbook supports their language learning tasks [46,47]. Originally conceptualized within

**Table 1. Contents of the research instrument.**

| Constructs | Dimensions (First-order) | Questionnaire code | Items | Sources |
|---|---|---|---|---|
| Task-technology Fit (TTF) | – | TTF1 | The English digital textbook is very useful in helping me to complete the unit learning task. | [4] [46,47] |
| | | TTF2 | The English digital textbook is very helpful for me to complete the unit learning task. | |
| | | TTF3 | The English digital textbook makes it very easy to complete the unit learning task. | |
| | | TTF4 | In general, the digital English textbook meets my English learning needs. | |
| Positive Emotion (POE) | – | POE1 | I feel active when learning the digital textbook. | [48,49] |
| | | POE2 | I feel happy when learning the digital textbook. | |
| | | POE3 | I feel enthusiastic when learning the digital textbook. | |
| | | POE4 | I feel excited when learning the digital textbook. | |
| | | POE5 | I feel proud when learning the digital textbook. | |
| | | POE6 | I feel delighted when learning the digital textbook. | |
| Digital textbook usage outcomes (DTUO[a]) | Learning Effectiveness (LEE) | LEE1 | The English digital textbook has made a positive impact on my view of English learning. | [50,51] |
| | | LEE2 | The English digital textbook arouses my interest in English learning. | |
| | | LEE3 | The English digital textbook encourages me to devote myself to English learning. | |
| | Continuance Intention (COI) | COI1 | I will continuously use the digital English textbook in the future. | [12, 52] |
| | | COI2 | I will be more willing to use the digital English textbook in the future. | |
| | | COI3 | I will continuously employ the digital English textbook in the future and ramp up its utilization. | |
| | | COI4 | I highly recommend others to use the digital English textbook. | |

Note: Note: TTF, task–technology fit; POE, positive emotion; DTUO, digital textbook usage outcomes; LEE, learning effectiveness; COI, continuance intention.

[a]DTUO is operationalized as a second-order construct comprising LEE and COI.

information systems research by Goodhue and Thompson [4], TTF measures the alignment between technology functions and individual task requirements. In this study, we combined items from these established scales and refined them into a four-item scale. While most of the TTF measures use five items, we streamlined items to keep the measure appropriate and concise. The scale directly assessed the perceived utility and supportiveness of the digital textbook to accomplish specific learning tasks.

**Positive emotion (POE).** Positive emotion was assessed by a six-item scale adapted from Casaló et al. [48] and Pekrun et al. [49]. Three items, namely happy, excited, and delighted, were borrowed from Casaló et al. to represent general affective valence. Three items, namely active, enthusiastic, and proud, were generated from the theoretical framework by Pekrun et al. to represent academic context. This combination can better reflect both the hedonic pleasure of the technology and the achievement-related emotions of the learning process. The items were reworded to the sentence stem: "I feel [emotion] when learning the digital textbook."

**Digital textbook usage outcomes (DTUO).** In accordance with the hierarchy in structural equation modeling, digital textbook usage outcomes (DTUO) were measured as a second-order construct, which is jointly indicated by two specific first-order dimensions, namely learning effectiveness (LEE) and continuance intention (COI). Learning effectiveness was measured using a scale adapted from Liu et al. [50] and further contextualized to the English digital textbook setting. In this study, the dimension was conceptualized as learners' perceived behavioral and engagement-related effectiveness, an operationalization informed by the behavioral dimension of Kirkpatrick's evaluation model [51]. Given our focus on learners' experiences and engagement, this dimension was adopted to emphasize observable changes in learning attitudes, interest, and effort investment rather than objective learning outcomes. To measure continuance intention, we synthesized a four-item scale drawing on Bhattacherjee [12] and Roca et al. [52]. Although Bhattacherjee established the core construct within the broader Information Systems (IS) context, Roca et al. validated it specifically for e-learning systems, expanding the measure to include usage intensity and recommendation. Given our focus on digital textbooks, we adopted this e-learning approach to capture students' long-term commitment. Specifically, the items were adapted to assess both the intention to continue using the digital textbook and the willingness to recommend it to others. A detailed summary of all constructs, sources, and items is provided in Table 1 and S1 File.

## Data collection

The structure of this study was carefully designed to ensure the confidentiality and anonymity of all participants. Prior to data collection, ethical approval was obtained from the Institutional Review Board (IRB) of Qufu Normal University, where the corresponding author is affiliated (Approval No. 2025−087), dated June 3, 2025. The research adhered to the guidelines established by the IRB of Qufu Normal University and the Declaration of Helsinki issued by the World Medical Association. All participants provided informed consent electronically. Moreover, participants were free to withdraw from the study at any time, without the need to offer any reason for doing so.

Following authorization from relevant stakeholders, the research team launched a questionnaire survey on June 3, 2025, targeting undergraduate students enrolled in the College English course at Qufu Normal University. The survey was scheduled to run for one week, during which participants completed a validated and translated Chinese version of the questionnaire via an online platform (https://www.wjx.cn). The survey link was distributed to L2 learners by their English teachers through QQ. The questionnaire yielded a total of 292 responses. After excluding incomplete submissions and outliers, 259 valid responses (see S1 Data) were retained for analysis. Throughout the research process, the team strictly adhered to the ethical guidelines set by their home institution.

## Data analysis

This study implemented a two-step analysis in accordance with the guidelines of Anderson and Gerbing [53]. The initial phase involved performing confirmatory factor analysis (CFA) to evaluate reliability metrics, including Cronbach's alpha

and composite reliability (CR), in addition to assessing convergent and discriminant validity. The second was about structural equation modeling (SEM) analysis, which was used to see how well the model fit the data and to see if the path coefficients were significant. The study evaluated the overall model fit and the robustness of the path coefficients within the proposed framework with the maximum likelihood estimation approach. Descriptive statistics and Cronbach's alpha were calculated using SPSS 27.0, and the structural model was assessed with AMOS 31.0.

## Results

### Common method variance

To mitigate the possible common method variance (CMV), we implemented procedural and statistical remedies [54]. Procedurally, respondents were assured of anonymity and informed that their participation was voluntary, which helped reduce evaluation apprehension and social desirability bias. In addition, the questionnaire was designed to separate key constructs, and all items were phrased in a clear and neutral manner to minimize ambiguity and method-related bias.

Statistically, a preliminary single-factor confirmatory factor analysis (CFA) was conducted. The model fit did not fit well ($x^2$/df$=7.047$, AGFI$=0.535$, GFI$=0.638$, and RMSEA$=0.153$), indicating that a single factor did not account for most of the data variations.

To further examine CMV, we implemented the unmeasured latent method construct (ULMC) approach [55]. A common latent factor (CLF) was inserted into the measurement model, indicating loadings on all 17 indicators and uncorrelated with the theoretical constructs. Although the chi-square difference test was statistically significant ($\Delta x2(17)= 63.069$, $p < .001$), likely due to the sensitivity of this statistic to sample size, other fit indices remained stable. Specifically, the variations in CFI ($\Delta$CFI$=0.014$), TLI ($\Delta$TLI$=0.010$), and RMSEA ($\Delta$RMSEA$=0.008$) were marginal and fell within recommended practical ranges for model robustness [56]. Furthermore, the differences in standardized factor loadings between the original measurement model and the CLF model ranged from 0.000 to 0.073, which were well below the recommended cutoff of 0.200 [57], and all substantive loadings remained statistically significant. These findings indicate that CMV was unlikely to have affected the estimated relationships in the structural model.

### Descriptive statistics and correlation analysis

Table 2 presents the descriptive statistics and Pearson correlation coefficients for task–technology fit (TTF), positive emotion (POE), and digital textbook use outcomes (DTUO), including the mean (M), standard deviation (SD), skewness, and kurtosis values.

To assess data normality, the Shapiro–Wilk test was first conducted. Although the results were statistically significant ($p < .05$), suggesting deviations from strict normality, the Shapiro–Wilk test is known to be highly sensitive to minor departures from normality in large samples (N$=259$) [58]. Therefore, skewness and kurtosis were further evaluated following the criteria proposed by Kline [59], whereby absolute skewness values greater than 3.0 and kurtosis values greater than 10.0 indicate severe non-normality. In the present study, all of the values of skewness and kurtosis were within these

Table 2. Descriptive statistics and correlations between the variables.

| Variable | M | SD | Skewness | Kurtosis | TTF | POE | DTUO |
|---|---|---|---|---|---|---|---|
| TTF | 3.93 | 0.73 | −0.56 | 0.80 | 1 | | |
| POE | 3.57 | 0.74 | 0.11 | −0.06 | .612** | 1 | |
| DTUO | 3.90 | 0.71 | −0.78 | 1.54 | .755** | .681** | 1 |

Note: TTF, Task-technology fit; POE, Positive emotion; DTUO, Digital textbook usage outcomes. **$p < .01$.

limits, suggesting that our data did not violate the assumption of normality to a great extent and were appropriate for the subsequent analyses.

As shown in Table 2, the mean scores of the three constructs, measured on a five-point Likert scale, ranged from 3.57 to 3.93, with SDs between 0.71 and 0.74. TTF had the highest mean value, and POE showed the lowest. Pearson correlation analysis revealed that all of the constructs were significantly and positively related to each other ($p < .01$). In particular, TTF was significantly correlated with POE ($r = .612$, $p < .001$) and DTUO ($r = .755$, $p < .001$), and POE was also significantly related to DTUO ($r = .681$, $p < .001$). The significant correlations between all the constructs, as shown in the present study, were in the proposed theoretical direction and thus provided the initial evidence for the research hypotheses.

Moreover, as the inter-construct correlations were below the recommended threshold of 0.85 [59], our data structure is free from conceptual redundancy; therefore, it is appropriate for the subsequent use of structural equation modeling to test the causal paths.

## Measurement model assessment

The measurement model is assessed in terms of individual item reliability, internal consistency reliability, convergent validity, and discriminant validity (see Table 3).

The individual item reliability was assessed based on the standardized factor loadings of each item on its corresponding latent construct. Following established guidelines [44], loadings greater than 0.70 are generally considered adequate. The results of this study show that all items exhibit factor loadings surpassing the 0.70 benchmark, thereby confirming satisfactory reliability of the items within the measurement model, as detailed in Table 3. In addition, both sub-dimensions loaded strongly on the second-order construct DTUO, with factor loadings of 0.933 for learning effectiveness and 0.968 for continuance intention, supporting the specification of DTUO as a second-order construct.

Internal consistency reliability was evaluated using the Cronbach's alpha (α) and composite reliability (CR) [44]. In Table 3, all the constructs met the suggested criteria for internal consistency. To be specific, TTF (α = 0.907, CR = 0.909) and POE (α = 0.921, CR = 0.921) are above the suggested criteria. In addition, the higher-order construct, DTUO (α = 0.926, CR = 0.937), also meets the suggested criteria of internal consistency, which indicates the reliability of the combined indicator.

Convergent validity was tested by using the average variance extracted (AVE), with a minimum threshold of 0.5 [60]. The AVE of the constructs in Table 3 is above the suggested threshold. In particular, the second-order construct, DTUO, had an AVE of 0.681, which indicates that the construct has a good amount of convergent validity.

Discriminant validity was assessed based on the Fornell–Larcker criterion [60]. According to this criterion, the square root of AVE for each construct should be higher than the correlation between the construct and any other construct.

Table 3. Reliability and convergent validity of the constructs.

| Constructs | Dimensions (First-order) (β) | Items | Factor loading | Cronbach's α | CR | AVE |
|---|---|---|---|---|---|---|
| TTF | | TTF1–4 | 0.810 - 0.887 | 0.907 | 0.909 | 0.715 |
| POE | | POE1–6 | 0.748 - 0.885 | 0.921 | 0.921 | 0.661 |
| DTUO (Second-order) | LEE (β = 0.933) | LEE1–3 | 0.804 - 0.872 | 0.926 | 0.937 | 0.681 |
| | COI (β = 0.968) | COI1–4 | 0.765 - 0.840 | | | |

Note: TTF, task–technology fit; POE, positive emotion; DTUO, digital textbook usage outcomes; LEE, learning effectiveness; COI, continuance intention. β represents the standardized factor loadings from the second-order construct to its respective sub-dimensions.

In Table 4, the square roots of AVE for TTF (0.846), POE (0.813), and DTUO (0.825) were higher than the correlation between the constructs, indicating that the three latent variables have an adequate discriminant validity.

## Structural model analysis

**Assumptions of multivariate analysis.** Prior to the structural model assessment, assumptions of multivariate analysis such as independence of observations, multicollinearity, linearity, and statistical stability were examined. Independence of observations was ensured by the cross-sectional survey collection where each respondent's response were considered independent. Independence of residuals was further supported by a Durbin–Watson statistic of 1.810 for the regression model of the combined outcome (DTUO), which lies within the acceptable range of 1.5 to 2.5.

Multicollinearity was examined through the variance inflation factors (VIFs) of the two predictors (TTF and POE), both obtaining VIF = 1.598, well below the conservative threshold of 3.0 [44]. Therefore, we can assume that multicollinearity is unlikely to bias the parameter estimates. Linearity was assumed based on the model specified and is supported by the satisfactory overall model fit indices reported in the following section.

Statistical stability of parameter estimates was examined through bootstrapping the model with 5,000 resamples, as the robustness of the results can be concluded from the reported 95% bias-corrected confidence intervals in the hypothesis testing part.

**Assessment of model fit.** The overall fit of the model was evaluated using several well-established goodness- of-fit indices proposed by Hu and Bentler [61], including the ratio of chi-square to degrees of freedom ($\chi^2$/df, also reported as CMIN/DF), the comparative fit index (CFI), the Tucker-Lewis index (TLI), and the root mean square error of approximation (SRMR). As presented in Table 5, the structural model fit the empirical data perfectly. CMIN/DF (1.878) was below the criterion of 3. The absolute fit indices, such as RMSEA (0.058) and GFI (0.910), met the proposed criteria. In addition, the incremental fit indices (CFI = 0.970, TLI = 0.965, IFI = 0.971) and parsimony fit indices (PCFI = 0.821, PNFI = 0.794) all met

**Table 4. The discriminant validity of the measurement model.**

| Construct | TTF | POE | DTUO |
|---|---|---|---|
| TTF | **0.846** | | |
| POE | 0.653*** | **0.813** | |
| DTUO | 0.831*** | 0.754*** | **0.825** |

Note: TTF, Task-technology fit; POE, Positive emotion; DTUO, Digital textbook usage outcomes. Bold values on the diagonal are the square root of the Average Variance Extracted (AVE); off-diagonal values are the correlations between constructs. *** $p < 0.001$.

**Table 5. Results of model fit analysis.**

| Fitness Index | Criteria for Judgement | Metric | Fitting Situation |
|---|---|---|---|
| CMIN/DF | < 3 | 1.878 | Ideal |
| RMSEA | < 0.08 | 0.058 | Ideal |
| GFI | > 0.9 | 0.910 | Ideal |
| AGFI | > 0.8 | 0.880 | Ideal |
| IFI | > 0.9 | 0.971 | Ideal |
| TLI | > 0.9 | 0.965 | Ideal |
| CFI | > 0.9 | 0.970 | Ideal |
| PCFI | > 0.5 | 0.821 | Ideal |
| PNFI | > 0.5 | 0.794 | Ideal |

the psychometric thresholds. Taken together, the proposed theoretical model fit the empirical data well, and met the joint criteria for the acceptability of the model.

**Hypothesis testing.** The maximum likelihood method was used to estimate the structural model, and a bias-corrected bootstrap procedure (5,000 resamples) was used to test the significance of the mediation effects. Table 6 presents the standardized path coefficients and their 95% confidence intervals (CI).

First, the direct paths corresponding to H1, H2, and H3 were examined. The results indicate that task-technology fit (TTF) has a significant direct positive influence on digital textbook use outcomes (DTUO) ($\beta = 0.591$, $SE = 0.060$, $p < .001$), supporting H1. Additionally, TTF significantly and positively affects positive emotion (POE) ($\beta = 0.653$, $SE = 0.066$, $p < .001$), providing support for H2. POE was also found to exert a significant positive effect on DTUO ($\beta = 0.369$, $SE = 0.056$, $p < .001$), thereby supporting H3.

Second, the mediating role of positive emotion (H4) was examined by analyzing the indirect effect of TTF on DTUO. The results yielded a significant indirect effect ($\beta = 0.241$, $SE = 0.047$, $p < .001$). Crucially, the 95% bias-corrected bootstrap confidence interval [0.156, 0.341] excluded zero, confirming the significance of this mediation mechanism [59]. Furthermore, as the direct path from TTF to DTUO remained statistically significant after accounting for the mediator, positive emotion was identified as a significant partial mediator in the relationship, thereby providing strong support for H4.

Finally, the explanatory power of the structural model was evaluated using the coefficient of determination ($R^2$). As illustrated in Fig 2, the proposed model concurrently explains 42.6% of the variance in students' positive emotions and

**Table 6. Standardized direct, indirect, and total effects of the structural model.**

| Path | β | SE | 95% CI Lower | 95% CI Upper | p |
|---|---|---|---|---|---|
| **Direct effects** | | | | | |
| H1: TTF→DTUO | 0.591 | 0.060 | 0.449 | 0.722 | <.001 |
| H2: TTF→POE | 0.653 | 0.066 | 0.535 | 0.753 | <.001 |
| H3: POE→DTUO | 0.369 | 0.056 | 0.225 | 0.515 | <.001 |
| **Indirect effect** | | | | | |
| H4: TTF→POE→DTUO | 0.241 | 0.047 | 0.156 | 0.341 | <.001 |

Note. β = standardized coefficient; SE = standard error; CI = confidence interval. Indirect effects were tested using a bias-corrected bootstrap procedure with 5,000 resamples. TTF = task–technology fit; POE = positive emotion; DTUO = digital textbook usage outcomes.

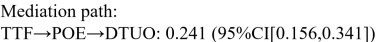

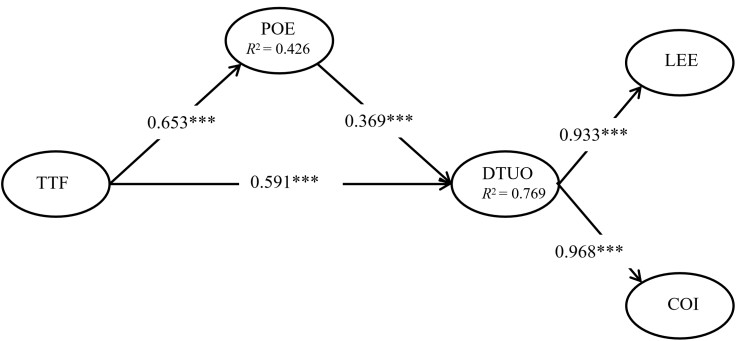

**Fig 2. The Structural model of TTF, POE, and DTUO of Chinese undergraduates.** Note: TTF = task–technology fit; POE = positive emotion; DTUO = digital textbook usage outcomes; N = 259; path coefficients are standardized estimates. *** $p < 0.001$. $R^2$ represents the variance explained.

76.9% of the variance in their digital textbook use outcomes ($R^2 = .426$ and .769, respectively). These findings suggest that the integration of task–technology fit and positive emotion within the S–O–R framework provides a substantial explanatory account for learners' success in digital textbook environments, demonstrating the model's high predictive relevance in this context.

## Discussion

This study examined a theoretically informed structural model specifying directional links among task–technology fit, positive emotion, and digital textbook use outcomes. Owing to the cross-sectional nature of the data, these relationships are understood as directional associations derived from model estimation rather than as evidence of causal effects. Accordingly, the results are interpreted as support for the hypothesized relational structure among the constructs, without implying causal ordering.

### Positive influence of task–technology fit on digital textbook usage outcomes

Our result demonstrates that TTF is positively associated with DTUO in the EFL environment, supporting H1. This is consistent with the fundamental assumption of TTF theory that the benefits of technology are fully realized when the functional affordances of a tool are closely matched to the needs of a pedagogical task [4,26,37]. This implies that the benefits of digital textbooks derive not from technology itself but from how accurately their intelligent functionalities align with the instruction process.

According to the Production-Oriented Approach, our results offer a further insight into the internal mechanism connecting TTF with DTUO. TTF acts as a functional bridge spanning across the three stages of POA: motivating, enabling, and assessing. In the motivating stage, the AI digital human "Ziyan" constructs relevant scenarios resonating with the communicative goals of the learners, effectively raising the psychological state for oral production. In the enabling stage, the intelligent portfolio and notebook enable the learners to retrieve stored linguistic resources, including target lexicon and text-based ideas, at the right timing needed for the production process. This "just-in-time" provision exemplifies a crucial shift from input to application, which is the behavioral effectiveness highlighted in the third level of Kirkpatrick's model [51]. Learners can also experience higher instructional effectiveness, which is reflected in their greater effort and proactive engagement in production.

In the final assessing phase, the functional fit of the feedback generated by AI enables the system to identify competence gaps in real time, and thus it reduces functional friction and provides individual diagnostic information at the right time. Consequently, the digital textbook can be interpreted as a "cognitive partner," supporting learners' engagement beyond content delivery. For this reason, DTUO can be considered a coherent, comprehensive answer in which the synchronization of task and technology during the POA process may guarantee not only the immediate learning success but also long-term psychological attachment to the digital learning environment.

### Positive influence of task-technology fit on positive emotion

In terms of the TTF–POE relation, the result indicates that a higher task–technology fit is significantly related to more positive affective experience in digital textbook learning (supporting H2). In line with the earlier findings, this suggests that a supportive technological environment directly boosts emotional involvement by promoting psychological well-being and continuous motivational involvement [29,40,62].

The positive emotional experience can be largely explained by the cognitive ease and psychological security afforded by the textbook's intelligent features. For example, the AI digital human "Ziyan" not only supports the oral practice tasks but also functions as a supportive learning partner. The technology provides instant encouragement and adaptive responses, which promote learners' feelings of social presence and confidence in communicative tasks. This is in line with the findings by Atuahene et al., who reported that digital multimodal affordances significantly enhanced learners' enthusiasm and positive psychological orientation towards the learning effort [63].

 

Likewise, the coordination of intelligent notebooks and portfolios creates a supportive interactive environment. When technology aligns with the learner's cognitive workflow, it is associated with functional fluency, which enhances the overall user experience and the intrinsic enjoyment of linguistic exploration [64]. This functional alignment is also linked to a state of perceived mastery, in which learners feel competent and adequately equipped to meet educational demands, supporting positive affect.

## Positive influence of positive emotion on digital textbook usage outcomes

Positive affect, in turn, predicts more favorable usage outcomes (supporting H3). This is consistent with the literature that learners' affective experiences during technology-mediated learning are strongly related to their perceived instructional value and continuance intention [30,32]. In the context of our digital textbook, positive affect experienced by the enjoyment and interest aroused by intelligent affordances, such as the embedded AI companion "Ziyan," not only directly enriches the experience of interactions in the digital textbook, but also reconstructs the learner's evaluation of the digital textbook utility.

Specifically, the affective engagement derived from the intelligent affordances is a psychological catalyst for learning effectiveness (LEE). Positive emotion during task engagement is associated with students' perception that their learning experience is valuable and meaningful. This is consistent with the view of Wang et al. [65], who argued that positive affective states are essential cues that the learning experience is of value and effectiveness. This also aligns with Guo [66], who found that positive affective states in e-learning environments have the role of enhancing perceived instructional gains.

Furthermore, the positive affect elicited has an obvious, strong stickiness factor that encourages continuance intention (COI), and Omar et al. have already offered a brilliant, well-supported description of this: intrinsic motivation arising from the experience of positive use is the main factor in determining a student's intention to adopt e-textbooks in the long run, and thus such emotional feedback directly and powerfully impacts learners' judgments about system acceptance [67]. Pushing this further, Chavali and Gundala (2022) point to an important and frequently neglected fact: perceived enjoyment is not a mere side effect, but a key driver that actually encourages learners to adopt digital resources as part of their study routine [62]. Therefore, POE is the natural adhesive that links instant gratification with long-term academic engagement, and also fits with the holistic view of the DTUO construct.

## Positive emotion as a mediator between task–technology fit and digital textbook usage outcomes

The confirmation of H4 holds that positive emotion is the crucial psychological intermediary that conveys the effect of task–technology fit to digital textbook use outcomes. This mediational role is consistent with the S–O–R framework, indicating that the effect of functional fit on usage outcomes is not just a mechanical process but one that is largely mediated and intensified through the learner's internal affective state [4,9]. Our findings are consistent with previous research indicating that affective processes are critical to the conversion of technological affordances to persistent academic engagement [29].

The mediating role can be explained by the mechanism by which intelligent features accommodate the functional requirements and emotional requirements [4,38]. In particular, while TTF reflects the extent to which tools such as intelligent notebooks and portfolios support the organization of linguistic resources, learners' emotional satisfaction with the availability of these resources is associated with a higher perceived value of the learning experience. This perceived competence appears to link the functional support provided by the tools with learners' sustained engagement and continued use of the digital textbook [37].

This mediation also suggests that perceived fit may facilitate the usage process, and positive emotion can support continued engagement. Functional adequacy alone may not be sufficient for engagement if the technology does not arouse learners' interest. As Joo et al. (2006) observed, affective feedback is the primary context through which learners evaluate functional benefits as personally relevant [68]. Through its friction-reducing, emotion-enhancing intelligent affordances, the

digital textbook ensures that objective task-technology alignment is successfully translated into evaluative response. This integrated pathway confirms the structural necessity of the S-O-R framework to explain the complexities of digital textbook use outcomes.

## Theoretical and pedagogical implications

**Theoretical implications.** This study offers empirical evidence on how task–technology fit (TTF) is associated with digital textbook use outcomes in the EFL context. In terms of situating the TTF perspective within the Stimulus–Organism–Response (S–O–R) framework, this study provides specific theoretical contributions that can add to the extant literature on both the functional and affective aspects of technology adoption.

First, this study situates the task–technology fit (TTF) perspective within the S–O–R framework, thereby providing an alternative to the direct-effect models often employed in prior studies [24]. Even though Jardina et al.'s previous research [24] was successful in establishing the functional relationship between fit and e-textbook use, our findings suggest that, in digital textbook learning settings, TTF is more appropriately considered as an environmental stimulus that positively predicts subsequent internal mind states. This perspective is consistent with observations by Yang and Jin that, for task-oriented technologies, it is the fit perception of the user who initiates the adoption behaviors [69]. By demonstrating that TTF influences internal affective states (H2), our study extends TTF theory by illustrating that it is not only TTF's direct effects on performance that matter, but also its ability to trigger the affective states that are prerequisite to the successful technology adoption.

Second, this study emphasizes the role of affective mechanisms in the S–O–R framework by proposing positive emotion as an important organismic state, which addresses Cheng's (2023) observation that previous research has often focused on the importance of cognitive evaluations instead of the importance of specific psychological constructs [70]. The abovementioned similarity between our findings and Chang's findings on m-learning supports the theoretical assumption that positive psychological states are a stable bridge in the adoption process across different digital mediums [71]. This implies that in the context of the digital textbook, the match between function and user needs is a precondition for emotional engagement. Therefore, this study goes beyond a purely utilitarian view of e-textbook adoption and indicates that usage success is ultimately based on the emotional satisfaction through the smooth match of tasks and technology.

Finally, this study further clarifies the conceptualization of "Response" (R) in the S-O-R framework by integrating learning effectiveness and continuance intention as one unified outcome construct. Despite recent integrated studies by Vafaei-Zadeh et al. tracing adoption intentions of the AI customer service sector [72], the learning nature of digital text-books requires a broader sense of success. Our findings show that in the educational context, a successful response is not merely the intention to use but also the interrelating synergy between the proximal outcome (effectiveness) and the distal outcome (continuance). The validation of hypothesis H3 indicates that two outcomes are co-dependent responses of the same affective mechanism. In this respect, this research proposes that theoretical models should account for the two-fold nature of success, treating immediate performance and future usage intentions as complementary aspects of effective technology integration.

**Practical implications.** The empirical results of this study provide concrete practical implications for the development of digital textbooks and educational practice emphasizing that long-term success is to a great degree contingent on the interaction of functional fit and emotional engagement.

First, digital textbook developers should consider ensuring alignment of structure between technological affordances and distinctive pedagogical tasks in order to potentially increase user outcomes. The results indicate that digital resources could benefit from moving beyond static digital counterparts to a model of functional customization. Such a model would include task-specific components, including dynamic concept mapping for knowledge construction or collaborative annotation for group inquiry, designed to support learners' cognitive processes. Moving from passive displays of content to more

 

active cognitive partners, these digital resources could reduce friction in operation and increase perceptions of usefulness. Such an approach would enable digital resources to be more than a repository of content, but instead facilitate the learning behaviors associated with academic success.

Second, interface designers should base their development on affective design strategies that generate positive affect. Because of the mediating role of affect we have uncovered in this study, an intuitively supportive environment is a strategic imperative for translating early interest into the continued engagement demonstrated by our results. In concrete terms, this might include designing distraction-free interfaces and adding subtle gamification to provide learners with instant rewarding feedback. Design efforts of this nature can guarantee that the technology not only assists with the cognitive demands of the task but also enhances the learner's overall well-being.

Finally, our findings highlight that the benefits of digital textbooks are fully harnessed only when pedagogies are deliberately and strategically matched with technological capabilities for teachers and educational organizations. Instructors are urged to reconceptualize learning tasks to exploit the unique affordances of the medium, such as interactive multimedia, using online data search tools, rather than simply seeing digital interfaces as electronic analogs of print media. Teachers can effectively engage the positive affect mechanisms uncovered in this study by ensuring the assigned tasks are intrinsically matched to the specific affordances of the technology. Such a teaching design generates the self-reinforcing cycle of engagement that will, in turn, increase the perceived effectiveness of learning and the persistence of digital textbook use.

## Limitations and future directions

Although this study obtained some results in theory and practice, there are still some limitations, and future studies can be further extended and improved in the following ways.

Firstly, the samples of this study are EFL learners in mainland China, which can illuminate the current digital textbook use trends to some extent but have geographical limits in culture, policy, and technology use environment. In the future, we can add samples from other countries or regions to carry out cross-cultural comparisons. In addition, the research in this paper adopted a cross-sectional design. Therefore, it failed to explain how the relationship between TTF, emotion, and use outcomes changes over time. Future research can adopt a longitudinal design to investigate how the POE changes and affects the digital textbook use outcomes.

Second, assessment of LEE was based on a relatively short, three-item, self-report scale. Although this operationalization is conceptually consistent with the behavioral dimension (Level 3) of the Kirkpatrick model, it primarily captures learners' perceived changes in learning attitudes, interest, and effort investment, rather than objective academic achievement. As such, it is vulnerable to self-report bias and limited construct breadth. Future research should evaluate it with objective indicators, such as course grades, quiz results, and usage logs, to ensure its stability in the digital textbook learning context.

Third, only positive emotion was considered as a mediating mechanism, so the study of affective states of the learners was rather limited in valence. Such an approach points well to the motivational drivers of adoption, but does not rule out the possible effect of negative affect, such as technostress or frustration. Accordingly, a future study might take a dual-valence approach to determine whether the fit of the task-technology interface was indeed more of a creator of positive affect or more of an inhibitor of negative affect. In addition, the use of retrospective reports does not allow for the granularity offered by contextualized emotion reports. While future adaptive textbooks could theoretically benefit from emotion-recognition techniques for facilitating learning, such technological advances must carefully address the looming ethical challenges. The issues of data privacy, informed consent, and algorithmic transparency are critical to guarantee the autonomy of the students and the strict ethical standards required in an emotion-aware learning environment.

Finally, the limitation of this study is that it mainly considers general psychological mechanisms and does not consider the moderating effects of demographic or individual background variables. For instance, gender, grade, and prior

experience with digital learning tools may affect students' fit perception and emotion processing. Unfortunately, the sample size was insufficient for a robust multigroup analysis. Future studies with larger and more diverse samples could systematically examine these moderating effects to support the empirical development of personalized instructional interventions and differential design requirements for digital textbooks.

## Conclusion

This study examines the effect of Task–Technology Fit (TTF) on digital textbook use outcomes (DTUO) through the Lens of the Stimulus–Organism–Response (S–O–R) framework. The empirical findings show that TTF is a strong environmental stimulus that can positively influence learners' positive emotions and use outcomes. Positive emotion here is the core mediating variable that demonstrates that functional fit between task and technology induces psychological immersion, which, in turn, contributes to positive DTUO represented as a composite construct of perceived learning effectiveness (LEE) and continuance intention (COI).

Overall, the study clarifies the chain mechanism of "Task–Technology Fit → Positive Emotion → Use Outcomes" in the digital learning context. The findings confirm that digital textbook implementation is not solely based on the functionality of technology but on a chain mechanism of structural fit inducing positive affective states, which, in turn, ensures positive learning outcomes and continuance.

## Supporting information

**S1 File. Survey items.**
(DOCX)

**S1 Dataset. Dataset collected from the survey.**
(XLSX)

## Author contributions

**Conceptualization:** Ning Mi.

**Data curation:** Ning Mi, Cunying Fan.

**Methodology:** Ning Mi.

**Resources:** Ning Mi, Cunying Fan.

**Software:** Cunying Fan.

**Validation:** Ning Mi.

**Visualization:** Cunying Fan.

**Writing – original draft:** Ning Mi.

**Writing – review & editing:** Cunying Fan.

## Acknowledgments

The authors would like to thank all participants for their valuable contributions to this study.

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
