## [Decision Letter · Decision Letter 0]

13 Oct 2025

Dear Dr. Mi,

Thank you for submitting your manuscript to PLOS ONE. After careful consideration, we feel that it has merit but does not fully meet PLOS ONE’s publication criteria as it currently stands. Therefore, we invite you to submit a revised version of the manuscript that addresses the points raised during the review process.

We look forward to receiving your revised manuscript.

Kind regards,

Intakhab Khan

Academic Editor

PLOS ONE

Journal Requirements:

Reviewers' comments:

Reviewer's Responses to Questions

**Comments to the Author**

1. Is the manuscript technically sound, and do the data support the conclusions?

Reviewer #1: Yes

2. Has the statistical analysis been performed appropriately and rigorously?

Reviewer #1: Yes

3. Have the authors made all data underlying the findings in their manuscript fully available?

Reviewer #1: Yes

4. Is the manuscript presented in an intelligible fashion and written in standard English?

Reviewer #1: Yes

Reviewer #1: The manuscript analyses in detail the serial mediation of positive emotion and learning effectiveness on digital textbook continuance. The ethical review and consent procedures are described and methodologically relevant. There are no issues related to concerns of dual publication, research publication ethics or ethics in general. There is transparent reporting of data availability and funding disclosures. The work enhances understanding of task-technology fit in educational technology. The authors declare no competing interests. As presented, the manuscript is methodically sound. It will serve as a resource for researchers, educators, and policymakers interested in digital learning.

**Do you want your identity to be public for this peer review?** For information about this choice, including consent withdrawal, please see our Privacy Policy

Reviewer #1: No

---

## [Author Response · Author response to Decision Letter 1]

4 Nov 2025

We have addressed all minor editorial and formatting suggestions in the manuscript, as detailed in the Response to Reviewers document. No additional substantive changes were requested by the reviewer. We thank the reviewer for their positive assessment and constructive comments.

---

## [Decision Letter · Decision Letter 1]

12 Jan 2026

Dear Dr. Mi,

Thank you for submitting your manuscript to PLOS ONE. After careful consideration, we feel that it has merit but does not fully meet PLOS ONE’s publication criteria as it currently stands. Therefore, we invite you to submit a revised version of the manuscript that addresses the points raised during the review process.

https://journals.plos.org/plosone/s/submission-guidelines#loc-laboratory-protocols . Additionally, PLOS ONE offers an option for publishing peer-reviewed Lab Protocol articles, which describe protocols hosted on protocols.io. Read more information on sharing protocols at https://plos.org/protocols?utm_medium=editorial-email&utm_source=authorletters&utm_campaign=protocols .

We look forward to receiving your revised manuscript.

Kind regards,

Leo Delaric Manansala, Ph.D.

Academic Editor

PLOS One

Journal Requirements:

Reviewers' comments:

Reviewer's Responses to Questions

**Comments to the Author**

Reviewer #2: (No Response)

Reviewer #3: (No Response)

2. Is the manuscript technically sound, and do the data support the conclusions?

Reviewer #2: (No Response)

Reviewer #3: Partly

3. Has the statistical analysis been performed appropriately and rigorously?

Reviewer #2: (No Response)

Reviewer #3: N/A

4. Have the authors made all data underlying the findings in their manuscript fully available?

Reviewer #2: (No Response)

Reviewer #3: Yes

5. Is the manuscript presented in an intelligible fashion and written in standard English?

Reviewer #2: (No Response)

Reviewer #3: Yes

Reviewer #2: Journal: PLOS ONE

MS ID: PONE-D-25-33273R1

Title: Task-technology fit drives digital textbooks continuance: Serial mediation of positive emotion and learning effectiveness

Version: 2 Date: 02 January 2026

Reviewer's report:

Comment 1—Language and flow: Minor grammatical and typographical errors are present. A thorough proofread is recommended. Examples include:

• “student’ s” in line 12 (page 1).

• “the the” in H1 description.

• Lines 170-171 (page 6) and 197-198 (page 7) are repetitive.

• “LOE” instead of “LEE” in line 335 (page 13).

• Factor loading for LEE2 “0.972” in Table 2 instead of “0.87” in Figure 2.

• “TTF (AVE = 0.717” in line 340 (page 13) instead of “0.715” in Table 2.

• “LEL” in Table 5 path label.

Comment 2—Methods section: Justify the sample size and the sampling technique used, and discuss potential sources of bias.

Comment 3—Methods section: Provide more detail on scale adaptation (e.g., expert review, pilot testing) and include a table summarizing all constructs, items, and sources in the main text or supplement.

Comment 4—Results section: Explore more robust methods to test for CMV or acknowledge that the used test, while common, is a basic one, and that the use of self-report data from a single source is a study limitation.

Comment 5—Descriptive statistics and correlation analysis section: The authors' claim about Kline's thresholds is inaccurate. Correcting this statement in a final revision would improve the manuscript's scholarly precision. Authors are encouraged to support the results with a normality test.

Comment 6—Structure model analysis section: Discuss the independence of observations, multicollinearity, linearity, and statistical stability of the results.

Comment 7—Limitations: Acknowledge more explicitly in the limitations that the discriminant validity between LEE and COI is borderline and that the high correlation suggests these constructs are very closely linked in their context.

Reviewer #3: The study is cross-sectional; however, parts of the writing read causally (e.g., “TTF drives COI”). You do acknowledge cross-sectional limitations later, but the framing should be consistent throughout (title, abstract, discussion). I recommend:

Replace causal verbs (“drives,” “leads to,” “results in”) with associational language (“is associated with,” “predicts,” “is linked to”) unless you provide stronger causal identification.

Add one short paragraph in the Methods or Discussion opening clarifying that results support the proposed directional model but do not establish causality.

You used a single-factor CFA and conclude CMV is unlikely because the one-factor model fits poorly.

This is a very common first step, but reviewers often see it as insufficient on its own, especially when all constructs are self-reported in one survey.

Actionable improvements:

Add a marker variable approach or a latent method factor test in CFA/SEM (even as a robustness check).

At minimum, report procedural remedies more explicitly (e.g., psychological separation, anonymity wording, item randomisation, reducing evaluation apprehension), not only ethics/confidentiality.

You note that “the square root of AVE for LEE fell slightly below its correlation with a related construct,” and you conclude discriminant validity is “acceptable.”

Given the strong correlation between LEE and COI (reported as high), this needs more than a brief reassurance.

Actionable improvements:

Report HTMT (Henseler et al.) for discriminant validity. Many journals increasingly expect HTMT alongside Fornell–Larcker.

Consider whether COI and LEE are conceptually too close in your operationalisation (see next point); if so, revise measurement or acknowledge overlap and interpret cautiously.

LEE is measured using only three items (and appears behavioural-leaning by your description), which you also acknowledge as a limitation.

But this construct is central (largest direct effect on COI). With a brief scale, the model may be capturing “perceived progress” rather than a broader concept of effectiveness.

Actionable improvements:

Clarify exactly what LEE represents: perceived learning gains, behavioural engagement outcomes, or self-reported effectiveness.

If feasible, add at least one objective proxy (course grade, quiz performance, usage analytics) or justify why self-reported LEE is appropriate and what it can/cannot claim.

If you cannot add objective outcomes, expand the limitations with a sharper statement on self-report bias and measurement breadth.

The paper suggests “emotion recognition interfaces that dynamically adjust content difficulty based on real-time learner emotion levels.”

This is a very strong claim with privacy, feasibility, and ethics implications. In the current manuscript, it appears abruptly and is not grounded in your empirical design (you did not measure emotion recognition, biometric data, or adaptive systems).

Actionable improvements:

Reframe this as a future research direction or an optional advanced design concept, not a policy requirement.

If you keep it as an implication, you need a short cautionary note about data privacy, consent, transparency, and algorithmic limitations, and cite relevant literature.

Conduct a full reference audit: ensure every reference is real, correctly formatted, and has a valid DOI/URL where applicable.

Cite: Alhaji Modu Mustapha, Megat Aman Zahiri Megat Zakaria, Noraffandy Yahaya, Hassan Abuhassna*, Babakura Mamman, Alhaji Modu Isa, and Muhammad Alkali Kolo, "Students‘ Motivation and Effective Use of Self-regulated Learning on Learning Management System Moodle Environment in Higher Learning Institution in Nigeria," International Journal of Information and Education Technology vol. 13, no. 1, pp. 195-202, 2023.

**Do you want your identity to be public for this peer review?** For information about this choice, including consent withdrawal, please see our Privacy Policy

Reviewer #2: **Yes:** Mahmoud A. Abdel-Fattah

Reviewer #3: **Yes:** Hassan Abuhassna

---

## [Author Response · Author response to Decision Letter 2]

13 Feb 2026

Please refer to the uploaded 'Response to Reviewers' file for our detailed point-by-point responses to the reviewers' comments.

---

## [Editor Report · Decision Letter 2]

19 Feb 2026

Task–Technology Fit and Digital Textbook Usage Outcomes: The Mediating Role of Positive Emotion within an S–O–R Framework

PONE-D-25-33273R2

Dear Dr. Mi,

We’re pleased to inform you that your manuscript has been judged scientifically suitable for publication and will be formally accepted for publication once it meets all outstanding technical requirements.

Kind regards,

Leo Delaric Manansala, Ph.D.

Academic Editor

PLOS One
---

## [Editor Report · Acceptance letter]

PONE-D-25-33273R2

PLOS One

Dear Dr. Mi,

I'm pleased to inform you that your manuscript has been deemed suitable for publication in PLOS One. Congratulations! Your manuscript is now being handed over to our production team.

Kind regards,

on behalf of

Dr. Leo Delaric Manansala

Academic Editor

PLOS One